# Parcellating connectivity in spatial maps

Christopher Baldassano[1], Diane M. Beck[2] and Li Fei-Fei[1]

[1] Department of Computer Science, Stanford University, Stanford, CA, USA
[2] Beckman Institute and Department of Psychology, University of Illinois at Urbana-Champaign, Urbana, IL, USA

## ABSTRACT

A common goal in biological sciences is to model a complex web of connections using a small number of interacting units. We present a general approach for dividing up elements in a spatial map based on their connectivity properties, allowing for the discovery of local regions underlying large-scale connectivity matrices. Our method is specifically designed to respect spatial layout and identify locally-connected clusters, corresponding to plausible coherent units such as strings of adjacent DNA base pairs, subregions of the brain, animal communities, or geographic ecosystems. Instead of using approximate greedy clustering, our nonparametric Bayesian model infers a precise parcellation using collapsed Gibbs sampling. We utilize an infinite clustering prior that intrinsically incorporates spatial constraints, allowing the model to search directly in the space of spatially-coherent parcellations. After showing results on synthetic datasets, we apply our method to both functional and structural connectivity data from the human brain. We find that our parcellation is substantially more effective than previous approaches at summarizing the brain's connectivity structure using a small number of clusters, produces better generalization to individual subject data, and reveals functional parcels related to known retinotopic maps in visual cortex. Additionally, we demonstrate the generality of our method by applying the same model to human migration data within the United States. This analysis reveals that migration behavior is generally influenced by state borders, but also identifies regional communities which cut across state lines. Our parcellation approach has a wide range of potential applications in understanding the spatial structure of complex biological networks.

Corresponding author
Christopher Baldassano,
chrisb33@cs.stanford.edu

# INTRODUCTION

When studying biological systems at any scale, scientists are often interested not only in the properties of individual molecules, cells, or organisms, but also in the web of *connections* between these units. The rise of massive biological datasets has enabled us to measure these second-order interactions more accurately, in domains ranging from protein–protein interactions, to neural networks, to ecosystem food webs. We can often gain insight into the overall structure of a connectivity graph by grouping elements into clusters based on their connectivity properties. Many types of biological networks have been modeled in terms of interactions between a relatively small set of "modules" (*Barabási & Oltvai,*

**Figure 1** **Parcellating connectivity in spatial maps** Given a set of elements arranged on a spatial map (such as points within the human cortex) as well as the connectivity between each pair of elements, our method finds the best parcellation of the spatial map into connected clusters of elements that all have similar connectivity properties. Brain image by Patrick J. Lynch, licensed under CC BY 2.5.

*2004*; *Hartwell et al., 1999*), including protein–protein interactions (*Rives & Galitski, 2003*), metabolic networks (*Ravasz et al., 2002*), bacterial co-occurrence (*Freilich et al., 2010*), pollination networks (*Olesen et al., 2007*), and food webs (*Krause, Frank & Mason, 2003*). In fact, it has been proposed that modularity may be a necessary property for any network that must adapt and evolve over time, since it allows for reconfiguration (*Alon, 2003*; *Hartwell et al., 1999*). There are a large number of methods for clustering connectivity data, such as k-means (*Kim et al., 2010*; *Golland et al., 2008*; *Lee et al., 2012*), Gaussian mixture modeling (*Golland, Golland & Malach, 2007*), hierarchical clustering (*Mumford et al., 2010*; *Cordes et al., 2002*; *Gorbach et al., 2011*), normalized cut (*Van den Heuvel, Mandl & Hulshoff Pol, 2008*), infinite relational modeling (*Morup et al., 2010*), force-directed graph layout (*Crippa et al., 2011*), weighted stochastic block modeling (*Aicher, Jacobs & Clauset, 2014*), and self-organized mapping (*Mishra et al., 2014*; *Wiggins et al., 2011*).

The vast majority of these methods, however, ignore the fact that biological networks almost always have some underlying spatial structure. As described by Legendre and Fortin: "In nature, living beings are distributed neither uniformly nor at random. Rather, they are aggregated in patches, or they form gradients or other kinds of spatial structures...the spatio-temporal structuring of the physical environment induces a similar organization of living beings and of biological processes, spatially as well as temporally" (*Legendre & Fortin, 1989*). In many biological datasets, we therefore wish to constrain possible clustering solutions to consist of *spatially-contiguous parcels*. For example, when dividing a DNA sequence into protein-coding genes, we should enforce that the genes are contiguous sequences of base pairs. Similarly, if we want to identify brain regions that could correspond to local cortical modules, we need each discovered cluster to be a spatially-contiguous region. Without spatial information, the discovered clusters may be difficult to interpret; for example, clustering functional brain connectivity data without spatial information yields spatially-distributed clusters that confound local modularity and long-distance interactions (*Lee et al., 2012*).

The problem is thus to a parcellate a spatial map into local, contiguous modules such that all elements in a module have the same connectivity properties (Fig. 1). In this paper we present the first general solution to this problem, introducing a new generative probabilistic model to parcellate a spatial map into local regions with connectivity properties that are as uniform as possible. Scientific insights can be gained from both the clusterings themselves (which identify the local spatial sources of the interaction matrix)

as well as the connections between the parcels, which summarize the original complex connectivity matrix. Our method yields better results than other approaches such as greedy clustering, and can help to determine the correct number of parcels in a data-driven way.

One of the most challenging spatial parcellation problems is in the domain of neuroscience. Modern human neuroimaging methods can estimate billions of connections between different locations in the brain, with complex spatial structures that are highly nonuniform in size and shape. Correctly identifying the detailed boundaries between brain regions is critical for understanding distributed neural processing, since even small inaccuracies in parcellation can yield major errors in estimating network structure (*Smith et al., 2011*).

Obtaining a brain parcellation with spatially coherent clusters has been difficult, since it is unclear how to extend standard clustering methods to include the constraint that only adjacent elements should be clustered together. Biasing the connectivity matrix to encourage local solutions can produce local parcels in some situations (*Cheng & Fan, 2014*; *Tomassini et al., 2007*), or distributed clusters can be split into their connected components after clustering (*Abraham et al., 2013*), but these approximations will not necessarily find the best parcellation of the original connectivity matrix. It is also possible to add a Markov Random Field prior (such as the Ising model) onto a clustering model to encourage connected parcels (*Jbabdi, Woolrich & Behrens, 2009*; *Ryali et al., 2013*), but in practice this does not guarantee that clusters will be spatially connected (*Honnorat et al., 2014*).

Currently, finding spatially-connected parcels is often accomplished using agglomerative clustering (*Thirion et al., 2014*; *Heller et al., 2006*; *Blumensath et al., 2013*; *Moreno-Dominguez, Anwander & Knosche, 2014*), which iteratively merges neighboring elements based on similarity in their connectivity maps. There are a number of disadvantages to this approach; most critically, the solution is only a greedy approximation (only a single pass over the data is made, and merged elements are never unmerged), which as will be shown below can lead to poor parcellations when there is a high level of noise. Edge detection methods (*Cohen et al., 2008*; *Wig, Laumann & Petersen, 2014*; *Gordon et al., 2014*) define cluster boundaries based on sharp changes in connectivity properties, which are also sensitive to localized patches of noisy data. Spectral approaches such as normalized cut (*Craddock et al., 2012*) attempt to divide the spatial map into clusters by maximizing within-cluster similarity and between-cluster dissimilarity, but this approach has a strong bias to choose clusters that all have similar sizes (*Blumensath et al., 2013*). It is also possible to incorporate a star-convexity prior into an MRF to efficiently identify connected parcels (*Honnorat et al., 2014*). This approach, however, constrains clusters to be convex (in connectivity space); as will be shown below, our method finds structures in real datasets violating this assumption, such as nested regions in functional brain connectivity data. All of these methods require explicitly setting the specific number of desired clusters, and are optimizing a somewhat simpler objective function; they seek to maximize the similarity between the one-dimensional rows or columns of the connectivity matrix, while our method takes into account reordering of the both the rows and columns to make the between-parcel 2D connectivity matrix as simple as possible.

Our model is highly robust to noise, has no constraints on the potential sizes and shapes of brain regions, and makes many passes over the data to precisely identify region boundaries. We validate that our method outperforms previous approaches on synthetic datasets, and then show that we can more efficiently summarize both functional and structural brain connectivity data. Our parcellation of human cortex generalizes more effectively across subjects, and reveals new structure in the functional connectivity properties of visual cortex.

To demonstrate the wide applicability of our method, we apply the same model to find spatial patterns in human migration patterns within the United States. Despite the fact that this is an entirely different type of data at a different spatial scale, we are able to find new insights into how state borders shape migratory behavior. Our results on these diverse datasets suggest that our analysis could have a wide range of potential applications in understanding biological networks. It is also important to note that the "spatial adjacency" constraint of our method could also be used for other, nonspatial notions of adjacency; for example, clustering an organism's life into contiguous temporal segments based on its changing social interactions.

## MATERIALS AND METHODS

### Probabilistic model

Intuitively, we wish to find a parcellation $\mathbf{z}$ which identifies local regions, such that all elements in a region have the same connectivity "fingerprint." Specifically, for any two parcels $m$ and $n$, all pairwise connectivities between an element in parcel $m$ and an element in parcel $n$ should have a similar value. Our method uses the full distribution of all pairwise connectivities between two parcels, and finds a clustering for which this distribution is highly peaked. This makes our method much more robust than approaches which greedily merge similar clusters (*Thirion et al., 2014*; *Blumensath et al., 2013*) or define parcel edges where neighboring voxels differ (*Thirion et al., 2006*; *Wig, Laumann & Petersen, 2014*; *Gordon et al., 2014*). The goal of identifying modules with similar connectivity properties is conceptually similar to weighted stochastic block models (*Aicher, Jacobs & Clauset, 2014*), but it is unclear how these models could be extended to incorporate the spatial-connectivity constraint.

We would like to learn the number of regions automatically from data, and additionally impose the requirement that all regions must be spatially-connected. We can accomplish both goals more efficiently in a single framework, by using an infinite clustering prior on our parcellation $\mathbf{z}$ which simultaneously constrains regions to be spatially coherent and does not limit the number of possible clusters. Specifically, since the mere existence of a element (even with unknown connectivity properties) changes the spatial connectivity and thus affects the most likely clustering, we must employ a nonparametric prior which is *not marginally invariant*. Other Bayesian nonparametric models allow for spatial dependencies between datapoints, but the only class of CRPs which is not marginally invariant is the distance-dependent Chinese Restaurant Process (dd-CRP) (*Blei & Frazier, 2011*). Instead of directly sampling a label for each element, the dd-CRP prior assigns each element $i$ a

link to a neighboring element $c_i$. The actual parcel labels $\mathbf{z(c)}$ are then defined implicitly as the undirected connected components of the link graph. Intuitively, this allows for changes in the labels of many elements when a single connection $c_i$ is modified, since it may break apart or merge together two large connected sets of elements. Additionally, this construction allows the model to search freely in the space of parcel links $\mathbf{c}$, since every possible setting of the parcel links corresponds to a parcellation satisfying the spatial-coherence constraint.

Given a parcellation, we must then specify a generative model for the data matrix $\mathbf{D}$. Analogous to the approach taken in stochastic block modeling (*Aicher, Jacobs & Clauset, 2014*), we model the connectivity between each pair of parcels as a separate distribution with latent parameters. To allow efficient collapsed sampling (see below), we utilize a Normal distribution for each set of connectivities between parcels, and the conjugate prior for the latent parameters.

Mathematically, our generative clustering model is:

$$\mathbf{c} \sim \text{dd-CRP}(\alpha, f)$$
$$A_{mn}, \sigma^2_{mn} \sim \text{Normal-Inverse-}\chi^2(\mu_0, \kappa_0, \sigma^2_0, \nu_0)$$
$$D_{ij} \sim \text{Normal}(A_{\mathbf{z(c)}_i \mathbf{z(c)}_j}, \sigma^2_{\mathbf{z(c)}_i \mathbf{z(c)}_j}).$$

For $N$ elements and $K$ parcels: $\mathbf{c}$ is a vector of length $N$ which defines the cluster links for all elements (producing a region labeling vector $\mathbf{z(c)}$ of length $N$, taking values from 1 to $K$); $\alpha$ and $f$ are the scalar hyperparameter and $N \times N$ distance function defining the dd-CRP; $\mathbf{A}$ and $\boldsymbol{\sigma}^2$ are the $K \times K$ connectivity strength and variance between regions; $\mu_0$ and $\kappa_0$ are the scalar prior mean and precision for the connectivity strength; $\sigma^2_0$ and $\nu_0$ are the scalar prior mean and precision for the connectivity variance; and $\mathbf{D}$ is the $N \times N$ observed connectivity between individual elements.

The probability of choosing a particular $c_i$ in the dd-CRP is defined by a distance function $f$; we use $f_{ij} = 1$ if $i$ and $j$ are neighbors, and 0 otherwise, which guarantees that all clusters will be spatially connected. A hyperparameter $\alpha$ controls the probability that a voxel will choose to link to itself. Note that, due to our choice of distance function $f$, a random partition drawn from the dd-CRP can have many clusters even for $\alpha = 0$, since elements are only locally connected.

The connectivity strength $A_{mn}$ and variance $\sigma^2_{mn}$ between each pair of clusters $m$ and $n$ is given by a Normal-Inverse-$\chi^2$ (NI$\chi^2$) distribution, and the connectivity $D_{ij}$ between every element $i$ in one region and $j$ in the other is sampled based on this strength and variance. The conjugacy of the Normal-Inverse-$\chi^2$ and Normal distributions allows us to collapse over $A_{mn}$ and $\sigma^2_{mn}$ and sample only the clustering variables $c_i$. Empirically, we find that the only critical hyperparameter is the expected variance $\sigma^2_0$, with lower values encouraging parcels to be smaller (we set $\alpha = 10, \mu_0 = 0, \kappa_0 = 0.0001, \nu_0 = 1$ for all experiments).

To allow the comparison of hyperparameter values between problems with the same number of elements (e.g., the functional and structural datasets), we normalize the input matrix $D$ to have zero mean and unit variance. We then initialize the model using the Ward

clustering (see below) with the most likely number of clusters under our model, and setting the links **c** to form a random spanning tree within each cluster.

In summary, we have introduced a novel connectivity clustering model which (a) uses the full distribution of connectivity properties to define the parcellation likelihood, and (b) employs an infinite clustering model which automatically chooses the number of parcels and enforces that parcels be spatially-connected.

## Derivation of Gibbs sampling equations

To infer a maximum a posteriori (MAP) parcellation $\mathbf{z}$ based on the dd-CRP prior, we perform collapsed Gibbs sampling on the element links $\mathbf{c}$. A link $c_i$ for element $i$ is drawn from

$$p(c_i^{(new)}|\mathbf{c_{-i}}, D) \propto p(c_i^{(new)})p(D|\mathbf{z}(\mathbf{c_{-i}} \cup c_i^{(new)})) = p(c_i^{(new)})p(D|\mathbf{z}^{(new)})$$

$$\propto \begin{cases} \alpha & \text{if } c_i^{(new)} = i \\ 1 & \text{else} \end{cases} \prod_{k_1, k_2=1}^{|\mathbf{z}^{(new)}|} p(D_{z_{k_1}^{(new)}, z_{k_2}^{(new)}}). \tag{1}$$

To compare the likelihood term for different choices of $c_i^{(new)}$, we first remove the current link $c_i$, giving the induced partition $\mathbf{z}(\mathbf{c_{-i}})$ (which may split a region). If we resample $c_i$ to a self-loop or to a neighbor $j$ that does not join two regions, the likelihood term is based on the partition $\mathbf{z}(\mathbf{c_{-i}}) = \mathbf{z}$. Alternatively, $c_i$ can be resampled to a neighbor $j$ such that two regions $K'$ and $K''$ in $\mathbf{z}(\mathbf{c_{-i}})$ are merged into one region $K$ in $\mathbf{z}(\mathbf{c_{-i}} \cup c_i^{(new)}) = \hat{\mathbf{z}}$. Numbering the regions so that $z_i \in \{1 \cdots (K-1), K', K''\}$ and $\hat{z}_i \in \{1 \cdots (K-1), K\}$ gives

$$\frac{p(D|\hat{\mathbf{z}})}{p(D|\mathbf{z})} = \frac{\prod_{k=1}^{K} p(D_{\hat{z}_k, \hat{z}_K}) \prod_{k=1}^{K-1} p(D_{\hat{z}_K, \hat{z}_k})}{\prod_{k=1}^{K'} p(D_{z_k, z_{K'}}) \prod_{k=1}^{K''} p(D_{z_k, z_{K''}}) \prod_{k=1}^{K-1} p(D_{z_{K'}, z_k}) \prod_{k=1}^{K'} p(D_{z_{K''}, z_k})}. \tag{2}$$

Each term $p(D_{z_m, z_n})$ is a marginal likelihood of the NI$\chi^2$ distribution, which can be computed in closed form (*Murphy, 2007*):

$$p(D_{z_m, z_n}) = \frac{\Gamma(v_{mn}/2)}{\Gamma(v_0/2)} \left(\frac{\kappa_0}{\kappa_{mn}}\right)^{\frac{1}{2}} \frac{(v_0 \sigma_0^2)^{v_0/2}}{(v_{mn} \sigma_{mn}^2)^{v_{mn}/2}} (\pi)^{-n/2}$$

$$L = |z_m||z_n| \qquad \kappa_{mn} = \kappa_0 + L; \ v_{mn} = v_0 + L \qquad \mu_{mn} = \frac{\kappa_0 \mu_0 + L\bar{d}}{\kappa_{mn}}$$

$$\bar{d} = \frac{1}{L} \sum_{\substack{i \in z_m \\ j \in z_n}} D_{ij} \qquad s = \sum_{\substack{i \in z_m \\ j \in z_n}} (D_{ij} - \bar{d})^2 \qquad \sigma_{mn}^2 = \frac{1}{v_{mn}} \left(v_0 \sigma_0^2 + s + \frac{L\kappa_0}{\kappa_0 + L}(\mu_0 - \bar{d})^2\right).$$

Intuitively, Eq. (2) computes the probability of merging or splitting two regions at each step based on whether the connectivities between these regions' elements and the rest of the regions are better fit by one distribution or two.

In practice, the time-consuming portion of each sampling iteration is computing the sum of squared deviations $s$. This can be made more efficient by computing the $s$ values for the merged $\hat{\mathbf{z}}$ in closed form. Given that the connectivities $D_{K'} = \{D_{iK'}\}_{i \in k}$ between parcel $k$

and $K'$ have sum of squares deviations $s_{K'}$ and mean $\bar{d}_{K'}$, and similarly for $K''$, then the sum of squares $s_K$ for the connectivities between parcel $k$ and the merged parcel $K$ (merging $K'$ and $K''$) is:

$$
\begin{aligned}
s_K &= \sum_{d \in D_{K'} \cup D_{K''}} (d - \bar{d})^2 \\
&= \left( \sum_{d \in D_{K'} \cup D_{K''}} d^2 \right) - (|D_{K'}| + |D_{K''}|) \cdot \left( \frac{|D_{K'}| \cdot \bar{d}_{K'} + |D_{K''}| \cdot \bar{d}_{K''}}{|D_{K'}| + |D_{K''}|} \right)^2 \\
&= \left( \sum_{d \in D_{K'} \cup D_{K''}} d^2 \right) - \frac{|D_{K'}|^2}{|D_{K'}| + |D_{K''}|} \bar{d}_{K'}^2 - \frac{|D_{K''}|^2}{|D_{K'}| + |D_{K''}|} \bar{d}_{K''}^2 - 2 \frac{|D_{K'}||D_{K''}|}{|D_{K'}| + |D_{K''}|} \bar{d}_{K'} \bar{d}_{K''} \\
&= \left( \sum_{d \in D_{K'}} d^2 - |D_{K'}| \bar{d}_{K'}^2 \right) + \left( \sum_{d \in D_{K''}} d^2 - |D_{K''}| \bar{d}_{K''}^2 \right) \\
&\quad + \frac{|D_{K'}||D_{K''}|}{|D_{K'}| + |D_{K''}|} \left( \bar{d}_{K'}^2 + \bar{d}_{K''}^2 - 2 \bar{d}_{K'} \bar{d}_{K''} \right) \\
&= s_{K'} + s_{K''} + \frac{|D_{K'}||D_{K''}|}{|D_{K'}| + |D_{K''}|} (\bar{d}_{K'} - \bar{d}_{K''})^2.
\end{aligned}
$$

## Comparison methods

In order to evaluate the performance of our model, we compared our results to those of four existing methods. All of them require computing a dissimilarity measure between the connectivity patterns of elements $i$ and $j$. For a connectivity matrix $D$,

$$
W_{i,j} = \sqrt{ \sum_{a \neq i,j} (D_{i,a} - D_{j,a})^2 + \sum_{a \neq i,j} (D_{a,i} - D_{a,j})^2 }. \tag{3}
$$

"Local similarity" computes the edge dissimilarity $W_{i,j}$ between each pair of neighboring elements, and then removes all edges above a given threshold. Here we set the threshold in order to obtain a desired number of clusters. This type of edge-finding approach has been used extensively for neuroimaging parcellation (*Cohen et al., 2008*; *Wig, Laumann & Petersen, 2014*; *Gordon et al., 2014*). Additionally, this is equivalent to using a spectral clustering approach (*Thirion et al., 2006*) if clustering in the embedding space is performed using single-linkage hierarchical clustering.

"Normalized cut" computes the edge similarity $S_{i,j} = 1/W_{i,j}$ between each pair of neighboring elements, then runs the normalized cut algorithm of *Shi & Malik (2000)*. This draws partitions between elements $a$ and $b$ when their edge similarity $S_{a,b}$ is low relative to their similarities with other neighbors. Although computing the globally optimal normalized cut is NP-complete, an approximate solution can be found quickly by solving a generalized eigenvalue problem. This approach has been specifically applied to neuroimaging data (*Craddock et al., 2012*).

"Region growing" is based on the approach described in *Blumensath et al. (2013)*. First, a set of seed points is selected which have high similarity to all their neighbors,

since they are likely to be near the center of parcels. Seeds are then grown by iteratively adding neighboring elements with high similarity to the seed. Once every element has been assigned to a region, Ward clustering (see below) was used to cluster adjacent regions until the desired number of regions is reached.

"Ward clustering" requires computing $W_{i,j}$ between all pairs of elements (not just neighboring elements). Elements are each initialized as a separate cluster, and neighboring clusters are merged based on Ward's variance-minimizing linkage rule (*Ward, 1963*). This approach has been previously applied to neuroimaging data (*Thirion et al., 2014*; *Eickhoff et al., 2011*).

We also compared to random clusterings. Starting with each element in its own cluster, we iteratively picked a cluster uniformly at random and then merged it with a neighboring cluster (also picked uniformly at random from all neighbors). The process continued until the desired number of clusters remained.

## Synthetic data

To generate synthetic connectivity data, we created three different parcellation patterns on an 18 by 18 grid (see Fig. 2), with the number of regions $K = 5, 6, 9$. Each element of the $K \times K$ connectivity matrix $A$ was sampled from a standard normal distribution. For a given noise level $\sigma$, the connectivity value $D_{i,j}$ between element $i$ in cluster $\mathbf{z_i}$ and element $j$ in cluster $\mathbf{z_j}$ was sampled from a normal distribution with mean $A_{\mathbf{z_i},\mathbf{z_j}}$ and standard deviation $\sigma$. This data matrix was then input to our method with $\sigma_0^2 = 0.01$, which returned the MAP solution after 30 passes through the elements (approximately 10,000 steps). Both our method and all comparison methods were run for 20 different synthetic datasets for each noise level $\sigma$ and the results were averaged.

We also performed a supplementary experiment using a more challenging three-spiral dataset (*Chang & Yeung, 2008*). We generated the connectivity matrix as above, and defined elements to be spatially adjacent if they were consecutive along a spiral or adjacent between neighboring spirals. In addition to our standard initialization scheme using the Ward clustering with highest probability according to our model, we also considered initializations with fixed numbers of clusters derived from Ward clustering ($K = 2, 10$) or initializations in which the links $c$ were chosen are random. The $\sigma_0^2$ hyperparameter was set to 0.01 as above, and the MAP solution was returned after 100 passes (or 1,000 passes for the random initialization).

Parcellations were evaluated by calculating their normalized mutual information (NMI) with the ground truth labeling. We calculate NMI as in *Strehl & Ghosh (2002)*. This measure ranges from 0 to 1, and does not require any explicit "matching" between parcels. For $N$ total elements, if $\mathbf{z}$ assigns $n_h$ elements to cluster $h$, $\mathbf{z_{gt}}$ assigns $n_l^{gt}$ elements to cluster $l$, and $n_{h,l}$ elements are assigned to cluster $h$ by $\mathbf{z}$ and cluster $l$ by $\mathbf{z_{gt}}$, this is given by

$$\text{NMI}(\mathbf{z}, \mathbf{z_{gt}}) = \frac{I(\mathbf{z}, \mathbf{z_{gt}})}{\sqrt{H(\mathbf{z})H(\mathbf{z_{gt}})}} = \frac{\sum_h \sum_l n_{h,l} \log(N n_{h,l}/(n_h n_l^{gt}))}{\sqrt{\left(\sum_h n_h \log(n_h/N)\right)\left(\sum_l n_l^{gt} \log(n_l^{gt}/N)\right)}}. \quad (4)$$
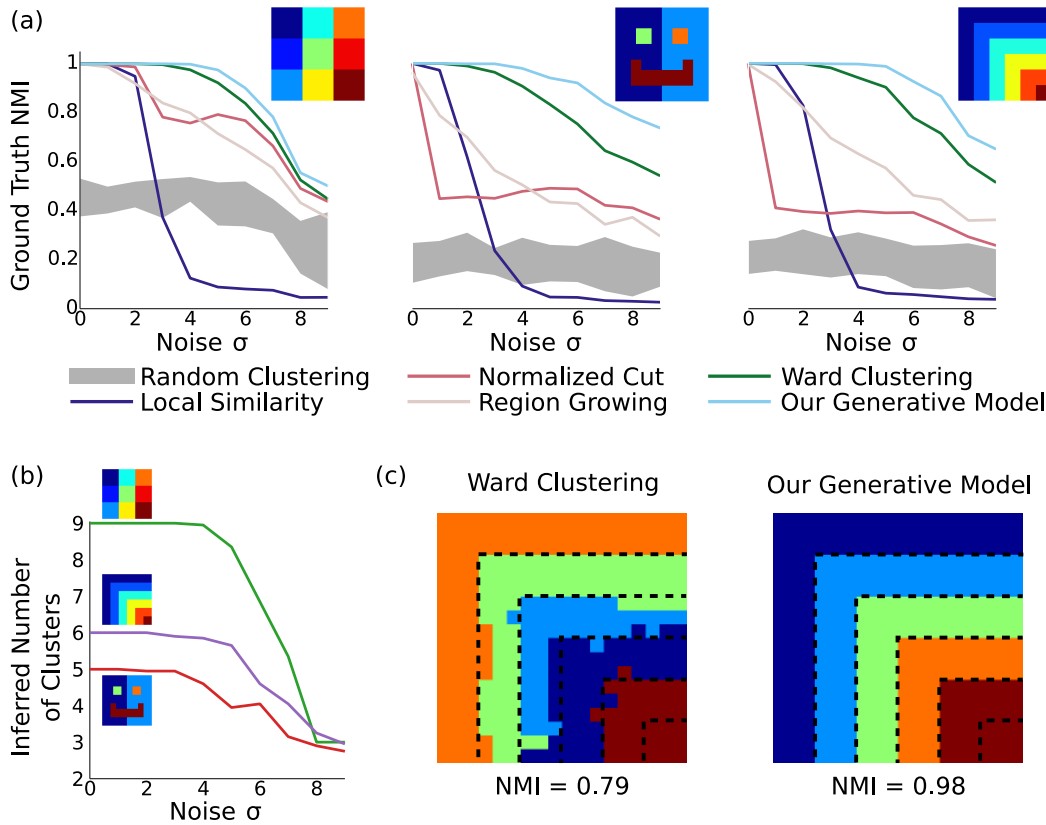

**Figure 2 Results on synthetic data.** (A) In three different synthetic datasets, our method is consistently better at recovering the ground-truth parcellation than alternative methods. This advantage is most pronounced when the parcels are arranged nonuniformly with unequal sizes, and the noise level is relatively high. Results are averaged across 20 random datasets for each noise level, and the gray region shows the standard deviation around random clusterings. (B) Our model can correctly infer the number of underlying clusters in the dataset for moderate levels of noise, and becomes more conservative about splitting elements into clusters as the noise level grows. (C) Example clusterings under the next-best clustering method and our model on the stripes dataset, for σ = 6. Although greedy clustering achieves a reasonable result, it is far noisier than the output of our method, which perfectly recovers the ground truth except for incorrectly merging the two smallest clusters.

## Human brain functional data

We utilized group-averaged resting-state functional MRI correlation data from 468 subjects, provided by the Human Connectome Project's 500 Subjects release (*Van Essen et al., 2013*). Using a specialized Siemens 3T "Connectome Skyra" scanner (Siemens AG, Berlin, Germany), data was collected during four 15-min runs, during which subjects fixated with their eyes open on a small cross-hair. A multiband sequence was used, allowing for acquisition of 2.0 mm isotropic voxels at a rate of 720 ms. Data for each subject was cleaned using motion regression and ICA + FIX denoising (*Smith et al., 2013*; *Salimi-Khorshidi et al., 2014*) and then combined across subjects using an approximate group-PCA method yielding the strongest 4,500 spatial eigenvectors (*Smith et al., 2014*). The symmetric 59,412 by 59,412 functional connectivity matrix $D_{a,b}$ was computed as the correlation between the 4,500-dimensional eigenmaps of voxels $a$ and $b$. For each

of $\sigma_0^2 = 2{,}000, 3{,}000, 4{,}000, 5{,}000$, we ran Gibbs Sampling for 10 passes (approximately 600,000 steps) to find the MAP solution. For comparison with individual subjects, we also computed functional connectivity matrices for the first 20 subjects with resting-state data in the 500 Subjects release.

The map of retinotopic regions in visual cortex was created by mapping the volume-based atlas from (*Wang et al., 2014*) onto the Human Connectome group-averaged surface.

### Human brain structural data

We obtained diffusion MRI data for 10 subjects from the Human Connectome Project's Q3 release (*Van Essen et al., 2013*). This data was collected on the specialized Skyra described above, using a multi-shell acquisition over 6 runs. Probabilistic tractgraphy was performed using FSL (*Jenkinson et al., 2012*), by estimating up to 3 crossing fibers with bedpostx (using gradient nonlinearities and a rician noise model) and then running probtrackx2 using the default parameters and distance correction. 2000 fibers were generated for each of the $1.7 \cdot 10^6$ white-matter voxels, yielding $3.4 \cdot 10^9$ total sampled tracks per subject (approximately 34 billion tracks in total). We assigned each of the endpoints to gray-matter voxels using the 32 k/hemisphere Conte69 registered standard mesh distributed for each subject, discarding the small number of tracks that did not have both endpoints in gray matter (e.g., cerebellar or spinal cord tracks). Since we are using distance correction, the weight of a track is set equal to its length. In order to account for imprecise tracking near the gray matter border, the weight of a track whose two endpoints are closest to voxels $a$ and $b$ is spread evenly across the connection between $a$ and $b$, the connections between $a$ and $b$'s neighbors, and the connections between $a$'s neighbors and $b$. Since the gray-matter mesh has a correspondence between subjects, we can compute the group-average number of tracks between every pair of voxels. Finally, since connectivity strengths are known to have a lognormal distribution (*Markov et al., 2014*), we define the symmetric 59,412 by 59,412 structural connectivity matrix $D_{a,b}$ as the log group-averaged weight between voxels $a$ and $b$. The hyperparameter $\sigma_0^2$ was set to 3,000, and Gibbs Sampling was run for 10 passes (approximately 600,000 steps) to find the MAP solution.

### Human migration data

We used the February 2014 release of the 2007–2011 county-to-county U.S. migration flows from the U.S. Census Bureau American Community Survey (*ACS*). This dataset includes estimates of the number of annual movers from every county to every other county, as well as population estimates for each county. We restricted our analysis to the continental U.S. To reduce the influence of noisy measurements from small counties, we preprocessed the dataset by iteratively merging the lowest-population county with its lowest-population neighbor (within the same state) until all regions contained at least 10,000 residents. This process produced 2,594 regions which we continue to refer to as "counties" for simplicity, though 306 cover multiple low-population counties. For visualization of counties and states, we utilized the KML Cartographic Boundary Files provided by the U.S. Census Bureau (*KML*).

One major issue with analyzing this migration data is that counties have widely varying populations (even after the preprocessing above), making it difficult to compare the absolute number of movers between counties. We correct for this by normalizing the migration flows relative to chance flows driven purely by population. If we assume a chance distribution in which a random mover is found to be moving from county $a$ to county $b$ based purely on population, then the normalized flow matrix is

$$D_{a,b} = \frac{M_{a,b}}{\left(\sum_{i,j} M_{i,j}\right) \cdot \frac{P_a P_b}{\left(\sum_i P_i\right)^2}} \quad (5)$$

where $M_{i,j}$ is the absolute number of movers from county $i$ to county $j$, and $P_i$ is the population of county $i$. This migration connectivity matrix $D$ is therefore a nonnegative, asymmetric matrix in which values less than 1 indicate below-chance migration, and values greater than 1 indicate above-chance migration. Setting $\sigma_0^2 = 10$, we ran Gibbs Sampling for 50 passes (approximately 130,000 steps) to find the MAP solution.

## RESULTS

### Comparison on synthetic data

In order to understand the properties of our model and quantitatively compare it to alternatives on a dataset with a known ground truth, we performed several experiments with synthetic datasets. We compared against random parcellations (in which elements were randomly merged together) as well as four existing methods: local similarity, which simply thresholds the similarities between pairwise elements (similar to (*Thirion et al., 2006*; *Cohen et al., 2008*; *Wig, Laumann & Petersen, 2014*; *Gordon et al., 2014*)); normalized cut (*Craddock et al., 2012*) which finds parcels maximizing within-cluster similarity and between-cluster difference; region growing (*Blumensath et al., 2013*), an agglomerative clustering method which selects stable points and iteratively merges similar elements; and Ward clustering (*Thirion et al., 2014*), an agglomerative clustering method which iteratively merges elements to minimize the total variance. Since these methods cannot automatically discover the number of clusters, they (and the random clustering) are set to use the same number of clusters as inferred by our method. We varied the noise level of the synthetic connectivity matrix from low to high, and evaluated the learned clusters using the normalized mutual information with the ground truth, which ranges from 0 to 1 (with 1 indicating perfect recovery).

As shown in Fig. 2, our method identifies parcels that best match the ground truth, across all three datasets and all noise levels. The naive local similarity approach performs very poorly under even mild noise conditions, and becomes worse than chance for high noise levels (for which most parcellations consist of single noisy voxels). Normalized cut is competitive only when the ground-truth parcels are equally sized (matching results from (*Blumensath et al., 2013*)), and is near-chance in the other cases. Region growing is more consistent across datasets, but does not reach the performance of Ward clustering,

which outperforms all methods other than ours. Our model correctly infers the number of clusters with moderate amounts of noise (using the same hyperparameters in all experiments), and finds near-perfect parcellations even at very high noise levels (see Fig. 2C).

We also evaluated our model on a three-spiral dataset previously used in clustering work (*Chang & Yeung, 2008*), showing that we outperform other methods regardless of initialization scheme (see Figure S1).

## Functional connectivity in the human brain

To investigate the spatial structure of functional connectivity in the human brain, we applied our model to data from the Human Connectome Project (*Van Essen et al., 2013*). Combining data from 468 subjects, this symmetric 59,412 by 59,412 matrix gives the correlation between fMRI timecourses of every pair of vertices on the surface of the brain (at 2 mm resolution) during a resting-state scan (in which subjects fixated on a blank screen). Using the anatomical surface models provided with the data, we defined vertices to be spatially adjacent if they were neighbors along the cortical surface.

Evaluating cortical parcellations is challenging since there is no clear ground truth for comparison, and different applications could require parcellations with different types of properties (e.g., optimizing for fitting individual subjects or for stability across subjects (*Thirion et al., 2014*)). One simple measure of an effective clustering is the fraction of variance in the full 3.5 billion element matrix which is captured by the connectivity between parcels (consisting of only tens of thousands of connections). As shown in Fig. 3A, our parcellation explains more variance for a given number of clusters than greedy Ward clustering; in order to achieve the same level of performance as our model, the simpler approach would require approximately 30 additional clusters. We can also measure how well this group-level parcellation (using data averaged from hundreds of subjects) fits the data from 20 individual subjects. Although the variance explained is substantially smaller for individual subjects, due both to higher noise levels and inter-subject connectivity differences, our model explains significantly more variance than Ward clustering with 140 clusters ($t_{19} = 2.97, p < 0.01$ one-tailed t-test), 155 clusters ($t_{19} = 3.67, p < 0.01$), or 172 clusters ($t_{19} = 1.77, p < 0.05$). The 220-cluster solutions from our model and Ward clustering generalize equally well, suggesting that our method's largest gains over greedy approximation occur in the more challenging regime of small numbers of clusters.

One part of the brain in which we do have prior knowledge about cortical organization is in visual cortex, which is segmented into well-known retinotopic field maps (*Wang et al., 2014*). We can qualitatively examine the match between our 172-cluster parcellation (Fig. 3C) and these retinotopic maps on an inflated cortical surface, shown in Fig. 3D. First, we observe a wide variety in the size and shape of the learned parcels, since the model places no explicit constraints on the clusters except that they must be spatially connected. We also see that we correctly infer very similar parcellations between hemispheres, despite the fact that bilateral symmetry is not enforced by the model. The earliest visual field maps (V1, V2, V3, hV4, LO1, LO2) all radiate out from a common representation of the fovea (*Brewer & Barton, 2012*), and in this region, our model generates ring parcellations

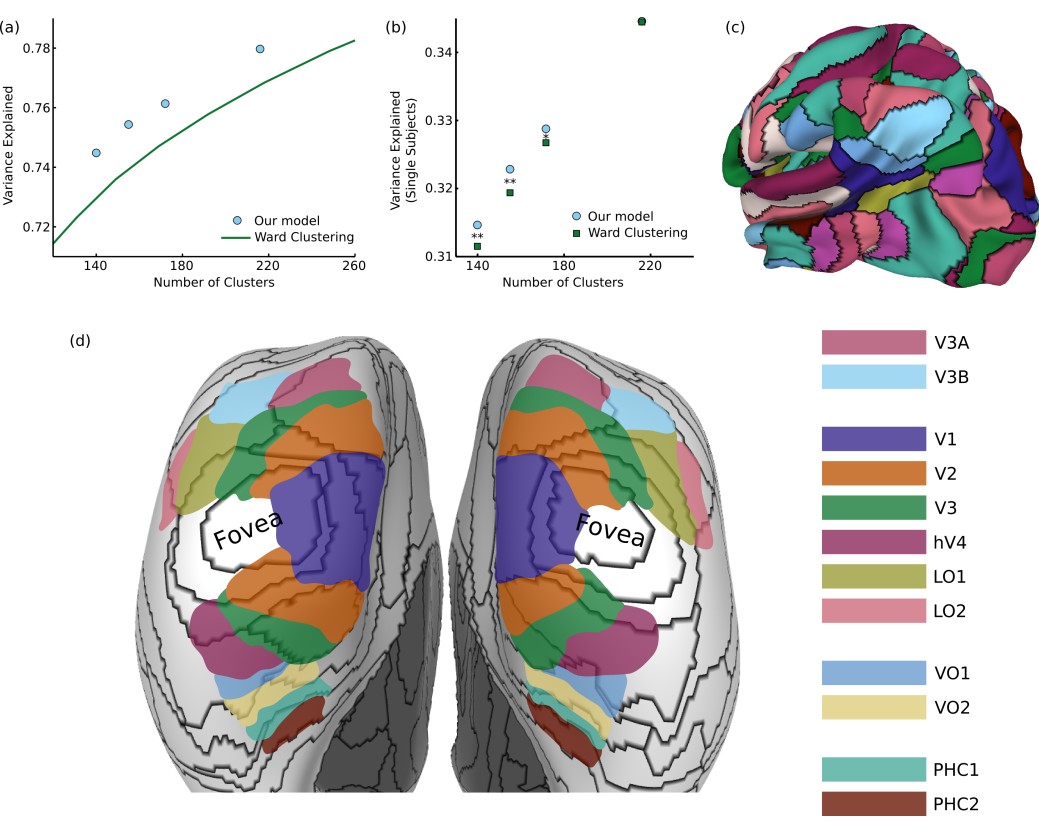

**Figure 3** **Results on functional brain connectivity** (A) Our model consistently provides a better fit to the data than greedy clustering, explaining the same amount of variance with 30 fewer clusters (different points were generated from different values of the hyperparameter $\sigma_0^2$). (B) When using our group-learned clustering to explain variance in 20 individual subjects, we consistently generalize better than the greedy clusters for cluster sizes less than 200 (* $p < 0.05$, ** $p < 0.01$). (C) A sample 172-cluster parcellation from our method. (D) Comparison between our parcels and retinotopic maps, showing a transition from eccentricity-based divisions to field map divisions.

which divide the visual field based on distance from the fovea. The parcellation also draws a sharp border between peripheral V1 and V2. In the dorsal V3A/V3B cluster, V3A and V3B are divided into separate parcels. In medial temporal regions, parcel borders show an approximate correspondence with known VO and PHC borders, with an especially close match along the PHC1–PHC2 border. Overall, we therefore see a transition from an eccentricity-based parcellation in the early visual cluster to a parcellation corresponding to known field maps in the later dorsal and ventral visual areas.

## Structural connectivity in the human brain

Based on diffusion MRI data from the Human Connectome Project (*Van Essen et al., 2013*), we used probabilistic tractography (*Behrens et al., 2007*) to generate estimates of the strength of the structural fiber connections between each pair of 2 mm gray-matter voxels. Approximately 34 billion tracts were sampled across 10 subjects, yielding a symmetric 59,412 by 59,412 matrix in which about two-thirds of the elements are non-zero. Applying our method to this matrix parcellates the brain into groups of voxels that all had the

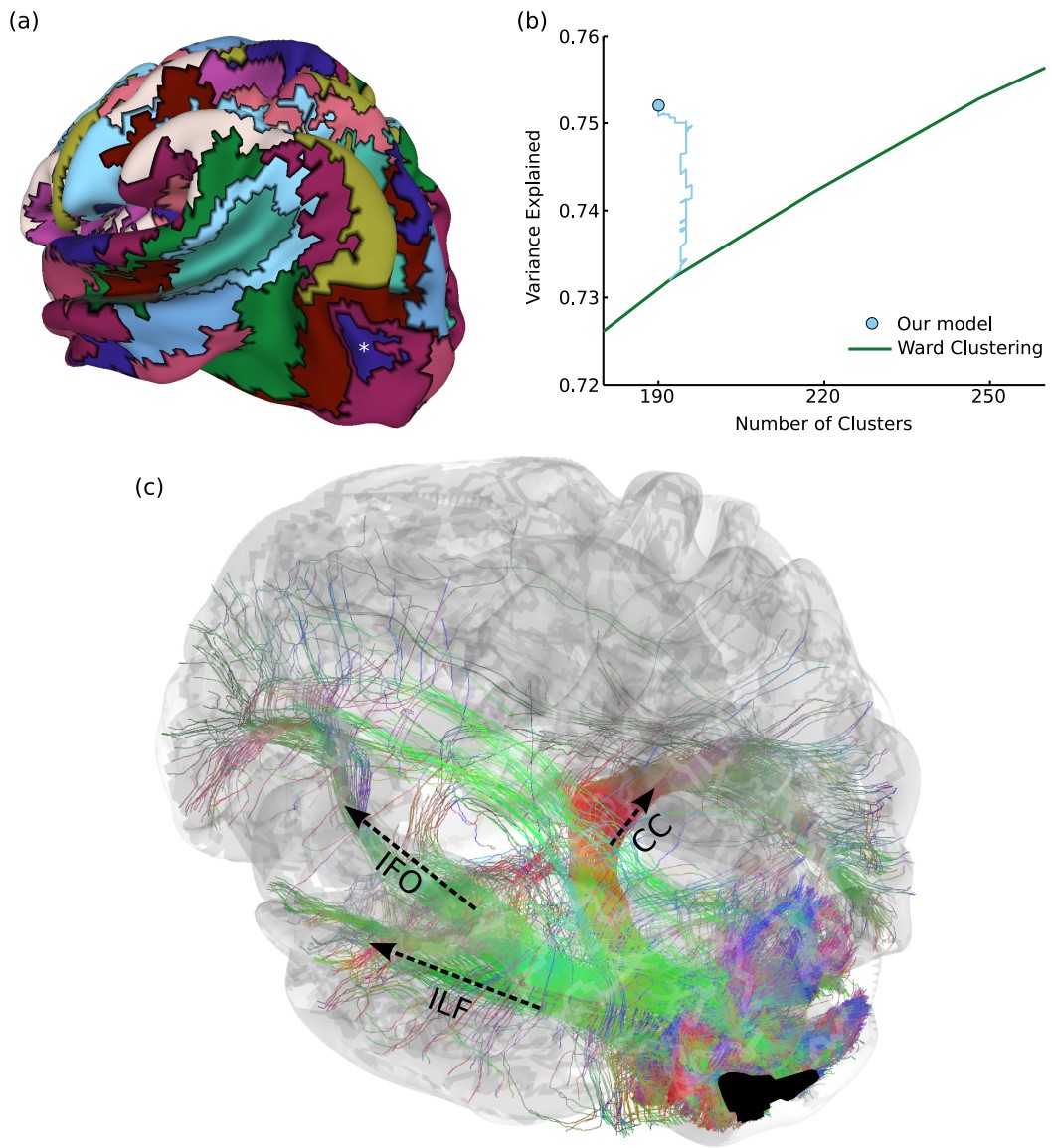

**Figure 4** **Results on structural brain connectivity.** (A) A 190-cluster parcellation of the brain based on structural tractography patterns. (B) This parcellation fits the data substantially better than greedy clustering, which would require an additional 55 clusters to explain the same amount of variance. The blue path shows how our model fit improves over the course of Gibbs sampling when initialized with the greedy solution. (C) An example of 35,000 tracks (from one subject) connected to a parcel in the lateral occipital sulcus, marked with an asterisk in (A). These include portions of major fascicles such as the inferior longitudinal fasciculus (ILF), inferior fronto-occipital fasciculus (IFO), and corpus callosum (CC).

same distribution of incident fibers. This problem is even more challenging than in the functional case, since this matrix is much less spatially smooth.

Figure 4A shows a 190-region parcellation. Our clustering outperforms greedy clustering by an even larger margin than with the functional data, explaining as much variance as a greedy parcellation with 55 additional clusters. Figure 4B also shows how

the model fit evolves over many rounds of Gibbs sampling, when initialized with the greedy solution. Since our method can flexibly explore different numbers of clusters, it is able (unlike a greedy method) to perform complex splitting and merging operations on the parcels. Qualitatively evaluating our parcellation is even more challenging than in the previous functional experiment, but we find that our parcels match the endpoints of major known tracts. For example, Fig. 4C shows 35,000 probabilistically-sampled tracts intersecting with a parcel in the left lateral occipital sulcus, which (in addition to many short-range fibers) connects to the temporal lobe through the inferior longitudinal fasciculus, to the frontal lobe through the inferior fronto-occipital fasciculus, and to homologous regions in the right hemisphere through the corpus callosum (*Wakana et al., 2004*). Note that the full connectivity matrix was constructed from a million times as many tracks as shown in this figure, in order to estimate the pairwise connectivity between every pair of gray-matter voxels.

## Human migration in the United States

Given our successful results on neuroimaging data, we then applied our method to an entirely distinct dataset: internal migration within the United States. Using our probabilistic model, we sought to summarize the (asymmetric) matrix of migration between US counties as flows between a smaller number of contiguous regions. The model is essentially searching for a parcellation such that all counties within a parcel have similar (in- and out-) migration patterns. Note that this is a challenging dataset for clustering analyses since the county-level migration matrix is extremely noisy and sparse, with only 3.8% of flows having a nonzero value.

As shown in Fig. 5A, we identify 83 regions defined by their migration properties. There are a number of interesting properties of this parcellation of the United States. Many clusters share borders with state borders, even though no information about the state membership of different counties was used during the parcellation. This alignment was substantially more prominent than when generating random 83-cluster parcellations, as shown in Fig. 5B. As described in the Discussion, this is consistent with previous work showing behavioral differences caused by state borders, providing the first evidence that state membership also has an impact on intranational migration patterns. Greedy clustering performs very poorly on this sparse, noisy matrix, producing many clusters containing only one or a small number of counties, and has a lower NMI with state borders than even the random parcellations.

The 10 most populous clusters (Fig. 5C) cover 18 of the 20 largest cities in the US, with the two largest parcels covering the Northeast and the west coast. Some clusters roughly align with states or groups of states, while other divide states (e.g., the urban centers of east Texas) or cut across multiple states (e.g., the "urban midwest" cluster consisting of Columbus, Detroit, and Chicago). As shown in Fig. 5D, our method succeeds in reordering the migration matrix to be composed of approximately piecewise constant blocks. In this case (and in many applications) the blocks along the main diagonal are most prominent, but this assortative structure is not enforced by the model. Though largely symmetric,

 

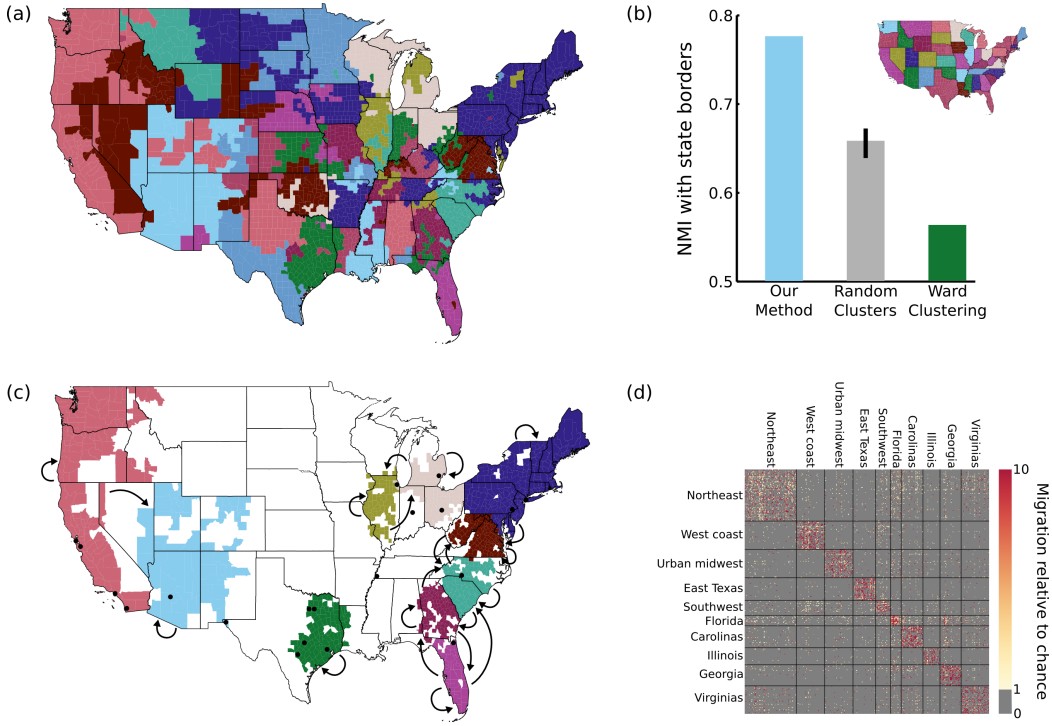

**Figure 5 Results on migration dataset.** (A) Our parcellation identified 83 contiguous regions within the continental US, such that migration between these regions summarizes the migration between all 2594 counties. (B) This parcellation was better aligned with state borders than an 83-cluster random parcellation (95% confidence interval shown) or an 83-cluster greedy Ward parcellation. (C) The top 10 clusters (by population) are shown, with arrows indicating above-chance flows between the clusters. The 20 most populous US cities are indicated with black dots for reference. (D) A portion of the migration matrix, showing the 1051 counties covered by the top 10 clusters.

some flows do show large asymmetries. For example, the two most asymmetrical flows by absolute difference are between the urban midwest and Illinois (out of Illinois = 1.3, into Illinois = 2.0), and Florida and Georgia (out of Georgia = 1.3, into Georgia = 2.0).

## DISCUSSION

In this work we have introduced a new generative nonparametric model for parcelling a spatial map based on connectivity information. After showing that our model outperforms existing baselines on synthetic data, we applied it to three distinct real-world datasets: functional brain connectivity, structural brain connectivity, and US migration. In each case our method showed improvements over the current state-of-the-art, and was able to capture hidden spatial patterns in the connectivity data. The gap between our approach and past work varied with the difficulty of the parcellation problem; hierarchical clustering would require ∼ 17% more clusters for the relatively smooth functional connectivity data and ∼ 29% more clusters for the more challenging structural connectivity data, and fails completely for the most noisy migration dataset.

Finding a connectivity-based parcellation of the brain's cortical surface has been an important goal in recent neuroimaging research, for two primary reasons. First, the shapes

and locations of connectivity-defined regions may help inform us about underlying modularity in cortex, providing a relatively hypothesis-free delineation of regions with distinct functional or structural properties. For example, connectivity clustering has been used to identify substructures in the posterior medial cortex (*Bzdok et al., 2014*), temporoparietal junction (*Mars et al., 2012*), medial frontal cortex (*Johansen-Berg et al., 2004*; *Kim et al., 2010*; *Crippa et al., 2011*; *Klein et al., 2007*), occipital lobes (*Thiebaut de Schotten et al., 2014*), frontal pole (*Moayedi et al., 2014*; *Liu et al., 2013*), lateral premotor cortex (*Tomassini et al., 2007*), lateral parietal cortex (*Mars et al., 2011*; *Ruschel et al., 2013*), amygdala (*Cheng & Fan, 2014*; *Mishra et al., 2014*), and insula (*Cauda et al., 2011*). Second, an accurate parcellation is necessary for performing higher-level analysis, such as analyzing distributed connectivity networks among parcels (*Power et al., 2013*; *Andrews-Hanna et al., 2010*; *Van den Heuvel & Sporns, 2013*), using connectivity as a clinical biomarker (*Castellanos et al., 2013*), or pooling voxel features for classification (*Xu, Zhen & Liu, 2010*). Consistent with our results, previous work has found that greedy Ward clustering generally fits the datasets best (in terms of variance explained) among these existing methods (*Thirion et al., 2014*).

Our finding of eccentricity-based resting-state parcels in early visual areas is consistent with previous results showing a foveal vs. peripheral division of visual regions based on connectivity (*Thomas Yeo et al., 2011*; *Lee et al., 2012*). Since our parcellation is much higher-resolution, we are able to observe nested clusters at multiple eccentricities. Our results are the first to suggest that higher-level retinotopic regions, especially PHC1 and PHC2, have borders that are related to changes in connectivity properties.

Parcellation based on structural tractography has generally been limited to specific regions of interest (*Mars et al., 2012*; *Johansen-Berg et al., 2004*; *Crippa et al., 2011*; *Klein et al., 2007*; *Thiebaut de Schotten et al., 2014*; *Moayedi et al., 2014*; *Liu et al., 2013*; *Tomassini et al., 2007*; *Mars et al., 2011*; *Ruschel et al., 2013*), in part due to the computational difficulties of computing and analyzing a full voxel-by-voxel connectivity matrix. Our parcellation for this modality is somewhat preliminary; probabilistic tractography algorithms are still in their infancy, with recent work showing that they produce many tracts that are not well-supported by the underlying diffusion data (*Pestilli et al., 2014*) and are of questionable anatomical accuracy (*Thomas et al., 2014*). As diffusion imaging and tractography methods continue to improve, the input connectivity matrix to our method will become higher quality and allow for more precise parcellation.

There has been detailed scientific study of both inter- and intra-national migration patterns for over a century, beginning with the 1885 work of *Ravenstein (1885)*. Even in this initial study (within the UK), it was clear that migration properties varied with spatial location; for example, rural areas showed large out-migration, while metropolitan areas showed greater in-migration, including long-distance migrants. The impact of state borders on migration behavior has not, to our knowledge, been specifically addressed, but there is a growing literature documenting differences in behaviors across state lines. Neighboring counties across state lines are less politically similar than those within a state, suggesting that a state border "creates a barrier to, or contains, political and economic

institutions, policies, and possibly movement" (*Tam Cho & Nicley, 2008*). State borders also play a role in isolating communities economically; this phenomenon gained a great of attention after Wolf's 2000 study (*Wolf, 2000*), showing that trade was markedly lower between states than within states (controlling for distance using a gravity model). Our results demonstrate in a hypothesis-free way that migration behavior is influenced by state identities, since our method discovers a parcellation related in many regions to state borders, without being given any information about the state membership of each county. Our results also show that state borders alone are not sufficient to capture the complexities of migration behavior, since other factors can override state identities to create other types of communities (such as in our "Urban midwest" parcel).

Since our algorithm makes many passes over the dataset, it does take longer than previous methods to find the most likely clustering. There are a number of possible approaches for speeding up inference which could be explored in future work. One possibility is to parallelize inference by performing Gibbs sampling on multiple elements simultaneously; although this would no longer be guaranteed to converge to the true posterior distribution, in practice this may not be an issue. Another option is to compute the Gibbs sampling probabilities only approximately (*Korattikara, Chen & Welling, 2014*), by using only a random subset of connectivities in a large matrix to approximate the likelihood of a proposed parcellation. It also may be possible to increase the performance of our algorithm even further by starting with many different initializations and selecting the solution with highest MAP probability.

## CONCLUSIONS

In summary, we have proposed the first general-purpose probabilistic model to intrinsically incorporate spatial information in its clustering prior, allowing us to search directly in the space of contiguous parcellations using collapsed Gibbs sampling. Our approach is far more flexible and precise than previous work, with no constraints on the sizes and shapes of the learned parcels. This makes our model more resilient to noise in synthetic tests, and provides better fits to real-world data drawn from three different domains. This diverse set of results suggests that our model could be applied to a large set of biological network datasets to reveal fine-grained structure in spatial maps. We have publicly released both MATLAB and python implementations of our method at http://goo.gl/xys4xh under a BSD open-source licence.

## ACKNOWLEDGEMENTS

Data were provided in part by the Human Connectome Project, WU-Minn Consortium (Principal Investigators: David Van Essen and Kamil Ugurbil; 1U54MH091657) funded by the 16 NIH Institutes and Centers that support the NIH Blueprint for Neuroscience Research; and by the McDonnell Center for Systems Neuroscience at Washington University.

Thank you Henry Jung for porting our MATLAB code to python, to Mike Arcaro for providing the map of retinotopic regions in visual cortex, and to Michelle Greene for reviewing early versions of this draft.

### Funding

Funding was provided by a National Science Foundation Graduate Research Fellowship under grant number DGE-0645962, and the Office of Naval Research Multidisciplinary University Research Initiative grant number N000141410671. The funders had no role in study design, data collection and analysis, decision to publish, or preparation of the manuscript.

### Grant Disclosures

The following grant information was disclosed by the authors:
National Science Foundation Graduate Research Fellowship: DGE-0645962.
Office of Naval Research Multidisciplinary University Research Initiative: N000141410671.

### Competing Interests

The authors declare there are no competing interests.

### Author Contributions

- Christopher Baldassano conceived and designed the experiments, performed the experiments, analyzed the data, contributed reagents/materials/analysis tools, wrote the paper, prepared figures and/or tables, reviewed drafts of the paper.
- Diane M. Beck and Li Fei-Fei conceived and designed the experiments, wrote the paper, reviewed drafts of the paper.

### Supplemental Information

Supplemental information for this article can be found online at http://dx.doi.org/10.7717/peerj.784#supplemental-information.

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
