# Peer review of "Parcellating connectivity in spatial maps"

_PeerJ, doi:10.7717/peerj.784_

## Round 0.1 · original submission · Minor Revisions

This is a solid work. Please make minor revisions based on reviewers' comments and provide a point by point response letter to explain the changes made.

Reviewer 1 ·

Basic reporting

No comments

Experimental design

No Comments

Validity of the findings

No comments

Additional comments

Authors have proposed a novel methodology which can learn connectivities with underlying spatial structure in consideration. The manuscript is well written and the idea is clearly explained. With that said, I have a minor comment as follows:
1. In line 155, authors have mentioned the Ward clustering method was used to provide an initial value to the number of clusters for the algorithm. It would be very useful to rerun the algorithm using a few other initial values to see how sensitive the algorithm is for different starting value.

Reviewer 2 ·

Basic reporting

Summary:
A new method is proposed to discover the locally connected clusters with respect to the spatial layout. The method is based on the Bayesian Nonparametric model: distance-dependent Chinese Restaurant Process, which have the advantages that the number of clusters can be discovered by Gibbs sampling. Thorough simulations and real data experiments are conducted, which demonstrates the superiority of the method.

Experimental design

thorough and solid

Validity of the findings

solid

Additional comments

Comments:
1) It is a well written paper. Authors explain the ideas very clearly and naturally. And the numerical experiments are sufficient and solid.
2) The biggest concern is that this paper seems just an straightforward application of existing method, there are no methodological or algorithmically novelty presented.
3) The authors should explain their methodological advantages if it is not just re-inventing the previous ideas.

Reviewer 3 ·

Basic reporting

No comments

Experimental design

For synthetic datasets, I suggest to try more challenging ones, such as shape sets: http://cs.uef.fi/sipu/datasets/

Validity of the findings

Make sense

Additional comments

This paper proposed a new model designed for locally-connected clusters. The model is interesting. My comments are followed:
i) For synthetic datasets, I suggest to try more challenging ones, such as shape sets: http://cs.uef.fi/sipu/datasets/
ii) Please make the codes publicly available
iii) There are some typos such as Sect. Synthetic Data 18x18... Please check the draft again.

---

## Round 0.2 · accepted · Accept

The revision fully address reviewers' concern and the manuscript has been greatly improved after revision. I suggest its acceptance now.